# Investigation of the Microstructure of Ti6Al4V Alloy by Coaxial Double Laser Metal-Wire Deposition

**DOI:** 10.3390/ma15227985

**Published:** 2022-11-11

**Authors:** Junjie He, Ryosuke Yokota, Yuji Imamiya, Keiichi Noriyama, Hiroyuki Sasahara

**Affiliations:** 1Department of Mechanical Systems Engineering, Tokyo University of Agriculture and Technology, 2-24-16, Koganei, Tokyo 184-8588, Japan; 2Nidec Machine Tool Corporation, 130 Roku-jizo, Ritto 520-3017, Japan

**Keywords:** laser metal wire deposition, Ti6Al4V alloy, microstructure observation, heat-affected zone band, simulation

## Abstract

Laser metal-wire deposition (LMwD) exhibits a larger molten pool and layer height during printing, compared to powder bed fusion additive manufacturing; in the present study, these features revealed a more inhomogeneous but easily observable microstructure. The coaxial double laser used herein makes the energy distribution of the molten pool more complex than that afforded by a single laser source, and the microstructure of the LMwD parts was more heterogeneous as well. We observed the microstructure of Ti6Al4V by the double LMwD as-built samples by conducting a laboratory experiment and a simulation. The precipitated martensite (α’) phase was defined after eliminating the influence of the β element in an X-ray diffraction analysis, which has not been discussed previously in the literature. We also propose a theory regarding the formation of heat-affected zone (HAZ) bands in an environment that includes the α’ phase. Our experiments revealed only white HAZ bands, which can be attributed to the solute partitioning caused by sequential thermal cycling and the absence of the β element. The microhardness of the HAZ band areas was lower than that of both the upper and lower sides. The simulation results indicate that the maximum temperature of 2925 °C restrains the generating of HAZ bands in the final two deposited layers, due to its great difference from the β transus temperature. Moreover, the higher heat accumulation in the upper layers promoted the migration of β-grain boundaries, which may explain why the coarse columnar β grains tended to grow at the edge area in the layers deposited later. We also observed that with the use of high temperature, the nucleation of β grains is more easily promoted in the lower layers. We conclude that the concentration of residual stress in the fusion zone and the first layer is favorable to the nucleation of equiaxed grains.

## 1. Introduction

Several additive manufacturing (AM) technologies are currently used in the aerospace, marine, and medical fields. The concept underlying these AM technologies is similar to the use of a heat source to fabricate the expected near-net-shaped parts [1]. In terms of the raw materials used, AM can be categorized into non-metal AM and metal AM. The most commonly applied AM technologies for metal AM are powder-bed fusion (PBF) and directed energy deposition (DED), which are used in accordance with the type and feeding method of the material [2,3]. Based on the heat source, AM types can be classified mainly as laser, electron-beam and arc-based AM [4,5]. The PBF and DED techniques have unique advantages that are not offered by conventional subtractive manufacturing [6,7]. Compared to PBF, laser metal wire deposition (LMwD) consists of a laser heat source and wire raw material; it has played a significant role in AM technologies due to its higher material utilization rate, larger printing size, excellent shaping efficiency, and lower material cost, although with worse deposit formation and accuracy, oxidation, and high residual stress [8,9]. All the inherent properties of LMwD have made it a potent and irreplaceable AM technology that continues to attract attention [10,11].

As a dual-phase (α+β) titanium alloy, Ti6Al4V is extensively applied in not only the aerospace industry but also in the medical field, due to its excellent inherent properties: superior weldability, low density, good anti-corrosive status, and high specific strength [12,13]. For the purposes of laser printing, Ti6Al4V alloy can form the martensite (α’) phase, due to the high undercooling from the laser heat source of approx. 10^4−5^ Ks^−1^, which is higher than the lowest value required for forming the α’ phase (400 Ks^−1^). Under these conditions, many α’ phases, filled in the coarse columnar β phase in as-built samples, are generally observed. However, the microstructures between layers are hard to observe and are still unclear now. Samodurova et al. compared the microstructure of titanium alloy by selective laser melting (SLM) and directed metal deposition; the results indicated that it is difficult to observe the microstructure of each layer of SLM due to the thinner layer thickness, whereas the microstructure of each layer deposited by directed metal deposition is easier to observe and investigate because of the larger molten pool and greater layer height compared to those from SLM [14]; this conclusion also applies to LMwD, which is helpful in evaluating the properties not only of laser metal-wire-deposited parts but also of PBF parts.

Moreover, such high undercooling also causes critical residual stress. As of this writing, a post-heat treatment, such as an annealing treatment, is the most effective method to reduce the residual stress after laser printing [12,15]. However, it would be better to reduce the residual stress during the deposition process directly; it is expected that the residual stress in the stress-relieved as-built parts can be reduced further after a post-heat treatment. Several research groups have explored this possibility; for example, Chen et al. [16] and Li et al. [17] investigated the effect of the deposition path on reducing residual stress during deposition, while Akbari and Kovacevic [7] considered the different inter-layer time intervals and Liu et al. [18] observed the substrate residual stress in a simulation. We are particularly interested in the findings reported by Lu et al., who studied residual stress by conducting simulations and in-situ experiments to verify the validity of the simulation results [19]. This research has shown that (i) the residual stress tends to concentrate in the first layer, and (ii) residual stress can be affected by adjusting the size, heat flux, and stiffness of the substrate. Although it was also demonstrated in their simulation that rounding off the sharp angles where the deposit was attached to the substrate helps reduce the residual stress, this method cannot be utilized in a real-world setting. Plenty of methods have been proposed that have attempted to reduce the residual stress of LMwD; nevertheless, the generation of high residual stress in the as-built parts seems inevitable since the high undercooling cannot be avoided. The issue of reducing residual stress merits further investigation.

Oxidation is another inevitable problem that must be addressed in the application of LMwD. Although a shielding gas system is part of every LMwD method, the use of both front and coaxial directions cannot prevent oxidation since the printing process takes place in atmospheric conditions. Fu et al. pioneered the use of underwater laser-metal-deposited Ti6Al4V alloy, in which an underwater environment protects the deposit from oxidation [20,21]. As a result, oxidation is improved, but the adverse effects of water reflection and refraction on the laser energy and the vaporized water used during the printing process cannot be avoided.

In addition to the residual stress and oxidation, uncontrollable mechanical properties are the third problem of LMwD [22]. The microstructure and chemical variations help to control the mechanical properties of as-built parts; however, the microstructure between layers is unclear and merits further study. The heat-affected zone (HAZ) band is an interesting microstructure between layers that is caused by the continuous heat treatment of depositions from post-layers, which was first discussed by Kobryn and Semiatin [9]. Kelly and Kampe proposed three possible hypotheses for HAZ band generation: oxidation, the segregation of alloying constituents, and the effect of thermal cycling [23,24]. Rigorous analyses eventually eliminated oxidation and the micro-segregation of alloying constituents as possible causes of HAZ bands; it has thus been concluded that the sequential heat treatment from upper-layer deposition causes the generation of HAZ bands, which conclusion is consistent with the proposal by Kobryn and Semiatin [9]. Ho et al. further investigated the HAZ bands observed with Ti6Al4V wire arc-based deposition, and their results confirmed the effect of α-β solute partitioning; in addition, the coarser dark HAZ band region, connected with a thinner white HAZ band region, was attributed to the coarsening of the transformation microstructure [25]. Although the generation principle of the HAZ band has been studied, its properties and influence on the as-built parts are not clear and merit further investigation.

In the present study, we carried out LMwD with an overlapping double laser heat source to study the microstructure of as-built parts. The first deposited layer of the multi-layer thin-walled part is special and differs from the deposition of subsequent layers, in that the first-layer deposit has higher undercooling and is in contact with the substrate; therefore, it is similar to a transition zone between the substrate and the subsequent layers. For this reason, we discuss the results concerning a single bead and the thin wall part separately. We provide a detailed elucidation of the microstructure difference and phase transformation, after describing the comparison of the as-built single bead parts and the thin wall part. The precipitated martensite phase, the regularities of the coarse columnar β phase, the changing of the phase transformation, and the HAZ band were investigated. We also conducted a simulation of temperature and residual stress distribution, to further clarify the microstructure characteristics.

## 2. Materials and Methods

### 2.1. Setup and Process Parameters

The LMwD single-bead and thin-wall specimens were deposited using an ytterbium fiber laser system (YLS-4500, IPG Photonics, Oxford, MA, USA) with two coaxially mounted irradiated laser torches [26,27] and a 1.0 mm diameter Ti6Al4V wire on the Ti6Al4V base metal. The printer was mounted on a six-axis delta frame structure, and the Repetier–Host software (to design and output G commands) was used to assist in the operation. To protect the molten pool and the deposited part from oxidizing, argon gas shielding nozzles were set coaxially, as shown in Figure 1.

The chemical composition of the Ti6Al4V wire and base metal is shown in Table 1. The process parameters during printing (Table 2) were optimized in previous experiments. Several parameters were tried, compared, and adjusted in terms of Equations (1) and (2), and the parameters that were applied in this study were selected based on the surface quality and microstructure observations of the as-built single-bead samples. With a laser power of 1000 W, a head feed speed of 500 mm/min, and a wire feed speed of 191 mm/min, a single bead of high surface quality was fabricated, as shown in Figure 2. The head height is set at 8 mm so that the spot of the double lasers could be overlapped, and the deposit can be melted successfully. The argon gas flow rate that was tested has the best efficiency at 20 L/min. The initial wire feed amount is set as 5 mm, to avoid deposit accumulation at the printing start point.

The parameters were designed and adjusted through Equations (1) and (2):(1)$$Q=\frac{60P}{F_{H}}$$
(2)$$S=\frac{10^{3}F_{W}\pi (\frac{D_{W}}{2})2}{F_{H}}$$
where *Q* is the heat input, *P* is the laser power, and F*_H_* is the head feed speed. *S* represents the cross-section area, *F_W_* is the wire feed speed, and *D_W_* represents the diameter of the laser spot. Z represents the head height of the substrate, L represents the argon gas flow rate, and E represents the initial wire feed amount.

### 2.2. Experiment Methods

The Ti6Al4V base metal (180 × 180 × 7 mm) was fixed on the stage by four clamps (one on each of the four corners). The printer head was then lowered into an appropriate position so that the head height between the torch and the substrate could be measured easily. Finally, the printer performed the G-code to achieve the intended deposit, as shown in Figure 2. During printing, the interval time between the two layers was set at 30 s. Regarding the thin-wall structure printing path, the printing direction was designed with a − Y and Y cycle and the orientation is shown in Figure 3b.

Melted Ti6Al4V wire is easily polluted by air conditioning during the printing process; during the process of cooling to room temperature, the deposited Ti6Al4V wire tends to absorb N_2_, O_2_, and H_2_ at 600 °C, 450 °C, and 300 °C, respectively [28]. In the present experiment, the shielding gas system was coaxially mounted with a laser torch, which provided limited near-spherical gas shielding. To avoid the contamination of air as much as possible in the deposited area, we used two methods simultaneously: (1) the head feed speed was controlled at an appropriate value so that the shielding gas area could cover and protect the deposited area for a longer time. However, the staying time of the laser torch cannot be extended because an excessively long pause would pose a higher possibility of the cooling metal being contaminated by the air environment. We tested the argon gas flow rate before this experiment and eventually set it at 20 L/min. (2) The G-code was designed to make the argon gas flow for an additional 1 sec at the end of the process so that the final deposited area would be well-protected, even when the printing process was completed.

The single-bead samples were deposited with a 50-mm gauge length, 2.65-mm width, and 1.2-mm thickness, as shown in Figure 3a. The single-pass seven-layer thin-wall component, with a 50-millimeter gauge length, 2.65-millimeter width, and 7.9-millimeter thickness, is shown in Figure 3b.

After the as-built single-bead and thin-wall samples were deposited, we cut both the single-bead and thin-wall samples up to obtain the cross-sections, in order to observe the inner microstructure. The cut samples were then ground using silicon carbide papers, with a series of grades from #320 to #2000, and polished with poly-crystalline diamond suspension. The prepared specimens were etched with Kroll’s reagent, consisting of 2 vol.% HF, 5 vol.% HNO_3_, and 43 vol.% H_2_O.

The micro-hardness of the cut thin-wall specimen was measured with a Vickers microhardness tester (HMV-G21DT, Shimadzu, Kyoto). To investigate the element composition, X-ray diffraction (XRD) (SmartLab, Rigaku, Tokyo) was performed, and the output was set as 45 kV and 200 mA. The macrostructure was captured via optical microscopy (VHX-6000, Keyence, Osaka, Japan), and the microstructure was observed using scanning electron microscopy (SEM) (VE8800, Keyence). The element composition was explored with energy-dispersive spectroscopy (EDS) (Ametek/Edax, Mahwah, NJ, USA). The acceleration voltage of the SEM and EDS was set at 20 kV. The temperature and residual stress distribution of the substrate were simulated and calculated using Simufact (Hamburg, Germany) software.

## 3. Results and Discussion

### 3.1. As-Built Samples’ Surface Evaluation

As shown in Figure 3a, the single-bead samples displayed smooth morphology, and the surface exhibited a silver color; a golden color was observed in the initial deposition area. These results indicated a high-quality surface with almost no contamination [29]. However, the microstructure of the outer edge area of single-bead exhibited a thin oxide layer as shown in Figure 4a, and the weird white points could also be discovered under EDS analysis, as shown in Figure 5. The results of Table 3 proved that oxidation exists in the edge area of the single-bead sample. Compared to the single-bead sample, the thin-wall as-built component shown in Figure 3b displayed worse oxidation resistance, according to the different surface qualities. During the printing process, its silver color changed to purple and blue on the second layer, then the purple color changed to the final gray color. The appearance of these colors could be attributed to the limited argon gas circulation system. In this printer, the argon gas nozzles were mounted coaxially near the torch, and the gas flowed from above the molten pool, which resulted in a limited gas-protected space. In addition, when the thin-wall part was fabricated, the gradually increasing height of the deposit hindered the normal circulation of the shielding gas around the molten pool and further reduced the gas protection. The surface morphology of the thin-wall part was also worse than that of the single-bead part. As the height increased, the width of the new layer tended to increase while the height tended to decrease.

### 3.2. Microstructure Analysis

Figure 4 shows the etched morphology of the cross-section of a typical single-bead Ti6Al4V sample in which the α’ phases, β phase, and α grain boundaries were observed. In terms of the differing microstructure, material source, and position, the single-bead track cross-section could be divided into three regions: a deposited zone (DZ), a fusion zone (FZ), and a HAZ [21]. In the DZ, the coarse columnar β grains can be observed directly, and many acicular martensite phases were filled in the β grains. The FZ, which is a transition area between the HAZ and the DZ, was filled with thinner equiaxial β-grains, compared to those in the DZ. The HAZ included many defects. The deposition angle, γ, was also measured as a parameter of the wettability of molten droplets on the substrate during the printing process.

Although Ti6Al4V is a dual-phase alloy that includes the α and β phases at room temperature, during the LMwD process, the α phase (hexagonal close-packed, (HCP)) can transform into the β phase (body-centered cubic, (BCC)) once the temperature has increased and exceeded the β transus temperature (T_β_, 980 °C). The transformation can happen further once the temperature increases to the liquid phase transition temperature (approx. 1655 °C) [20]. In this way, all the α phases are transformed into the liquid β phase. After the laser heat source passed, the temperature of the deposited area started to decrease rapidly. Since the transformation between the α and β phases is reversible, the liquid β phase has the tendency to transform into the α phase. Nevertheless, due to the relatively high cooling rate, instead of transforming into the α phase, the β phase transformed into the α’ phase after the temperature of the deposit cooled down to the martensite transformation temperature (M_S_). Thus, many parent columnar β grains are present (Figure 4), inside which many acicular α’ phases precipitated from the boundaries of β grains. Once the temperature fell below the M_S_, the secondary α’ (α’’) phase and tertiary α’ (α’’’) phase could be observed between the spacing of the prior acicular α’ phase; the observed phases are consistent with the study by Y. Fu et al. [20,21]. A detailed martensitic transformation process diagram is provided in the later discussions.

However, a 100% martensitic transformation cannot be achieved, and the transformation process is unstable, but the α phase may exist [30]. We also observed worse oxidation in the thin-wall part; as the α phase stabilizer, the oxygen element can increase the T_β_ and restrain the martensitic transformation process, to a certain extent. Thus, there was a higher probability that the α phase would be observed in the edge area of each layer, as shown in Figure 6d.

Another interesting phenomenon was the appearance of many overlapping columnar β grains in the single-bead sample; these can be ascribed to the special heat source and larger laser spot diameter in the present experiment. The laser spot diameter used in this study was 1.4 mm and was provided by a double irradiated laser head. In this way, the two combined and interacting laser spots provide a special heat source but not a normal Gaussian heat source [27]; that is, the center overlapping the ellipse zone of the laser spot has a maximum energy density, while the rest of the laser spot’s area maintains a normal energy density by a single laser, which resulted in a different and more complex energy gradient and microstructure than that afforded by a single laser source. With such a double-laser non-Gaussian heat source, the energy distribution in the molten pool during printing is not uniform and the β columnar grain distribution is inhomogeneous and complicated. Moreover, a larger laser spot with a diameter of 1.4 mm makes this phenomenon more complex. Many columnar β grains were thus observed. Moreover, although the single-bead parts showed a nearly silver color, indicating that there was no contamination (Figure 3a), oxidation was observed in the outer area using optical microscopy, with an average width of 0.1 mm (Figure 4).

The microstructure of a seven-layer thin-wall sample is shown in Figure 6a,e in different cutting orientations, i.e., the X–Z and Y–Z planes. Compared to the microstructure of the single-bead sample in Figure 4a, the microstructure of the thin-wall part exhibited worse oxidation, which can be observed in the edge area of each layer. The number and size of the pores of the thin wall part were decreased, compared to the single-bead sample. This may be a result of the upper layers’ re-heating. Due to the sequential thermal cycling, the pores of the lower layers became smaller and even disappeared.

Figure 6a,b shows the β grains with different colors, and the β grains marked with the numbers 1, 2, and 3 in Figure 6b displayed different depths of martensitic surface relief, macroscopically expressed as different etching reactions. The grain numbered 1 in the figure had the deepest color, which may be attributable to the different heat gradients. In the laser printing conducted in this study, the heat source that was combined with two irradiated lasers must have several heat gradients, which resulted in a complex heat gradient in the molten pool. As a result, some β grains showed a deeper color, while the outer areas displayed a shallower morphology. In addition, the continuous thermal cycling may be responsible for the different degrees of martensitic convexity, which is in agreement with the findings of J. Liu et al., who considered the effect of heat accumulation from the subsequent deposition [18], while the shallower martensite was caused by martensite decomposition.

The reason why β grains always grew from the inner center area to the outer area of the single-bead sample and the lower layers may also be related to the different heat gradients. Another probable reason for this grain growth orientation is the low thermal conductivity (7.9 W/mK) of Ti6Al4V alloy [20,21]. During the first layer’s printing, the heat in the deposit can be transferred to the substrate more quickly because of the higher undercooling and larger surface area of the base metal. With the increase in the layers, the temperature of the previous layer increased, which caused decreased undercooling. The heat had to transfer from the higher layer to the substrate layer (which had a small surface area), and then to the substrate. Both heat gradients and thermal conductivity may cause heat accumulation and may have resulted in this particular grain orientation.

The growth of β grains also showed some regularities in the thin-wall part [22,25]. The β grains tended to grow in the edge area of each layer, especially in the higher layers, as shown in Figure 6a marked with red arrows; the β grains were not observed in the center area of the higher layers, and it seemed that the β grains gathered at the edge areas in order to grow. This phenomenon may be attributed to the upper layers’ re-heating and heat accumulation behavior. During thin-wall structure-building, once the upper layer has begun to deposit, the lower layer undergoes the heat treatment from the laser heat source again and the top area of the substrate layer may even be re-melted, which would cause the disappearance of the β grains in the center area. However, the edge area underwent less-intense heat treatment, and the parent β grains remained and started to re-grow.

The accumulation of heat may have changed the β grain boundaries and resulted in the observed growth pattern [18]. Because of the upper layer’s re-heating, a HAZ band was generated in Figure 6a,b ande. Almost all the β grains in the upper layers were derived from the lower layers, and some coarse columnar β grains could grow, even through several layers (Figure 6e). In addition, the lower layers had more β grains. The lower the height, the more β grains were observed, which indicates that greater residual stress remained in the lower area. The aggregation of the equiaxed β grains may be a result of the high undercooling.

Observations by optical microscopy and SEM revealed a special phase (Figure 6c,d) that was distinguished from the α’ phase and other aspects of the microstructure. We thus conducted an XRD to observe the phase composition, as shown in Figure 7. In the XRD pattern, all peaks represented hexagonal crystals, indicating that all peaks represented the α/α’ phase [30]. The β phase either did not exist in the thin-wall sample or, alternatively, the content of the β phase was too low to be detected by the XRD system. With the influence of the β phase being eliminated, we speculate that this special phase is the precipitated α’ phase marked in Figure 6c,d. In addition, the element composition of the special phase area was almost the same as that of the martensite phase area shown by EDS.

Notably, we observed that the special phases grew with a mutually orthogonal grain orientation, which is a typical regularity of the α’ phase. The appearance of this type of α’ phase was also attributed to the re-heating. The α’ phase has unstable organization, and in this experiment, when the upper layers transferred heat to the lower layer, the unstable α’ phase decomposed into α and β phases, and the morphology of the precipitated α’ phase eventually became smaller and shallower [20,30]. Figure 6d shows the α’ phase, α’’ phase, and α’’’ phase.

HAZ bands were observed inside each layer (Figure 6a,e, indicated by the red arrows). According to a previously published work by Kobryn and Semiatin [9], the reason for this can be ascribed to successive re-heating from the upper layers’ laser printing, which generated a T_β_ position in the lower deposited layers. Kelly and Kampe performed a further study on the HAZ band and speculated that the generation of the HAZ bands is related to the coarsening of the α lamellar [23,24]. Ho et al. performed a more rigorous investigation of HAZ bands; their findings [25] verified the conclusions reported by Kobryn and Kelly. However, all these investigations of HAZ bands discussed the findings only within a microstructure environment of α and β phases (not including the α’ phase), and the diffusional transformation between the α and β phases was not applicable to the martensite transformation, which is a non-diffusion transition process. We have found no published study discussing the generation of HAZ bands when combined with a microstructure that is fully filled with the α’ phase. Our present experiment used double laser irradiation heads, which also resulted in a more complicated microstructure compared to those obtained with the other laser metal deposition (LMD) methods.

Thus, with the use of the present conditions, the α’ phase was present on both sides of the HAZ band and even inside the HAZ band, as shown in Figure 6b,c,f. During this printing process, when the lower layers were heated by the sequent deposition process, there was a parallel line position in the substrate layer, where the temperature rose to the T_β_ just in time, and the previous microstructure was mainly transformed into the β phase again first, as shown in Figure 8d. When the laser heat source had passed, the β phase typically transformed into the α’ phase by a non-diffusion transformation in terms of the high undercooling, and the HAZ band formed within the cooling process simultaneously, as shown in Figure 8e, which displays the microstructure changes to the thin wall in the upper layers.

The HAZ bands in the present experiment could be attributed to the constituent segregation as well [23,24,25]. The chemical variation between the phases during the cooling process resulted in solute partitioning. The element composition by EDS near the HAZ band is shown in Table 4 [31]. The element Al, which is an α phase stabilizer, exhibited differing compositions in the different regions [20,25]. However, since only white bands were observed, we suspected that the final micro-segregation was generated between the α’ phase and α phase, with no β phase. In addition, according to the XRD pattern results, only HCP crystal peaks were present, and no β phase was detected. The macroscopic etching reaction also indicated the presence of several thin white HAZ bands.

In the present thin-wall structure, a total of five HAZ bands were observed. There was also no HAZ band in the last two deposited layers (Figure 6a), which result is different from the findings reported by Kelly and Kampe [23,24] and Ho et al. [25], in which a HAZ band was not observed in the last 3–5 layers. Although we found no HAZ bands in the last two layers, the microstructure of this region was more homogeneous than that of the lower layers, which included the HAZ band. As shown in Figure 6a,c,e, the greater the number of layers, the fewer the β grains that were present. Especially in the last two layers, almost all the center black area was filled with the thinner, precipitated α’ phase, which is similar to the phase shown in Figure 6c. This result was different from a previous report [21], in which the top layer was filled with equiaxed β grains. The higher laser energy density in this study, which resulted in greater undercooling, may be responsible for the HAZ band and grain differences.

The average width of the HAZ band (Figure 6f) was 300 ± 20 μm, and the α’ phase in this region was coarser than that in other regions. A transition area was also discovered inside the HAZ band where no phase was observed, and the precipitated α’ phase was observed near this area (Figure 6f). Each HAZ band observed in this study exhibited a white color only, which is different from the findings described by Kelly and Kampe [23,24] and Ho et al. [25]. This difference in the HAZ bands’ appearance can be attributed to the solute partitioning. The available reheated β phase cooled down to room temperature rapidly, and the solute β phase was redistributed to the α’ phase and the α phase. As a result, the HAZ bands induced by the etching reaction showed a white color.

In addition, there are some differences in the height location among the HAZ bands in each layer, and different parameters, such as the head-feed speed, energy density, and the inappropriate stand distance between the laser torch and substrate layer, may be responsible for the differences in findings among these studies. However, this phenomenon also indicates that the position of HAZ bands could be affected, and we are conducting further experiments to investigate the factors involved.

### 3.3. Microhardness Analysis

Figure 9 shows the microhardness in the center area of the thin-wall part from the seventh layer to the substrate. The microhardness values in both the DZ and the FZ were higher than that in the substrate, and it was clear that the discovered martensite phase in the DZ and FZ was responsible for the hardness difference. The average hardness in area 2 was 376 HV, which is higher than that in area number 1, i.e., 366 HV. The reason for this could be ascribed to the different textures in the two areas. In Figure 6a, larger columnar β grains with many martensite phases can be observed in area 1, whereas in the upper layers the martensite was present only in the edge area; instead, precipitated martensite phases filled in the center area of the upper layers, which decreased the hardness. The hardness in the HAZ decreased gradually with the decrease in the distance from the substrate surface because of the gradually decreasing and eventually absent martensite phase. Moreover, the greater hardness in the FZ could be attributed to a large number of thinner equiaxed grains. According to the Hall–Petch mechanism, the more grains there are, the more grain boundaries there will be, which impedes the movement of dislocation and results in a higher hardness value [14].

We also measured the hardness of the HAZ band area. As indicated by the green arrows in Figure 7, the hardness values of the HAZ band area from the fifth layer to the first layer were 353, 372, 361, 362, and 343 HV, respectively. All five HAZ band areas showed slightly lower hardness values compared to that of both the upper and lower sides. This result is inconsistent with an investigation in which the hardness between the HAZ band area and both upper and lower sides showed no significant difference [23].

### 3.4. Simulation

To investigate the distribution of the temperature values and residual stress of the as-built parts during and after the above-described printing experiment, we performed a simulation with the Simufact software program. We designed a simulated single-bead track with a 50 mm gauge length of 2.65 mm in width and 1.2 mm in thickness at room temperature, and a simulated seven-layer thin-wall part with a 50 mm gauge length, 2.65 mm width, and 7.9 mm thickness. The size of both models was the same as those of the real parts.

For the printing of the single-bead part, a conventional Gaussian heat source was selected, with a power of 1000 W and a head speed of 500 mm/min [17]. For the thin-wall part, a volumetric laser heat source was used. In the actual experiment, two laser heat sources were provided, which caused a non-normal distribution of the temperature of the molten pool; in addition, the actual printing process was more complex compared to the simulations.

The mesh generation plays a significant role in the simulation, and to confirm the accuracy and efficiency (which are two critical standards in the calculation), we fabricated fine meshes around the deposition track, while the coarse meshes were designed outside of the track [18,19]. In this simulation process, the maximum size of the meshes was 2.88 × 2.88 × 1.75 mm, and the minimum size was 0.72 × 0.72 × 1.75 mm.

Figure 10a provides the temperature distribution in the substrate of the single-bead part deposition. As shown in Figure 10b, the shape of the molten pool was consistent with the actual molten pool of the as-built single-bead samples (Figure 4a), which indicates the appropriateness of the selected heat source [18]. The results also indicated that the parameters and the model of the heat source were reasonable. The maximum temperature reached 1897 °C, which is slightly higher than the liquid temperature of Ti6Al4V alloy (1655 °C, T_L_). Figure 10c also illustrates the significant stress aggregation in the deposition track, which indicates a high level of undercooling between DZ and FZ during the printing process. The concentrated residual stress may also help to explain the nucleation of β grains in the FZ, which will be discussed in detail in the following pages; the peak stress vector reached 842 MPa, which resulted in coolingshrinkage. The calculated distortion of the substrate was 0.16 mm [29].

The maximum temperature of the thin-wall simulation was 2925 °C (Figure 11a). Since there were six layers of thermal cycling, it was understandable that the peak temperature of the thin wall was higher than that of the single-bead part. This temperature is also high enough (higher than T_L_) that in such conditions, the melted deposit would be vaporized. With the help of the gasification recoil force, the generated gas had a strong influence on the molten pool, which caused an increase in grain nucleation, especially in the first layer, where the keyhole was larger than those of the other layers [18]. As a result, smaller β grains were observed in the lower layers. These results are in agreement with the conclusions that were based on the data in Figure 4 and Figure 6.

The heat gradients during the fifth layer’s printing process are displayed in Figure 11b. During the deposition of the fifth layer, the fourth and third layers showed a higher temperature gradient (red color), whereas the first and second layers showed a very low-temperature gradient. These phenomena indicated that: (1) the reason why HAZ bands were not observed in the last two layers could be ascribed to the higher temperature, which was far from T_β_. Ho et al. [25] also observed the phenomenon that the HAZ band is not discovered in the last three layers; however, the reason is not elucidated in their investigation. This conclusion could also explain the disappearance of the HAZ band in the top layers. It can also be inferred that a HAZ band is generated only when the thin wall deposits are more than three layers. (2) A high accumulation of heat was easier to achieve with a greater number of layers, as is shown in Figure 11a. In light of the low thermal conductivity of Ti6Al4V, the transfer of heat dissipation from the upper layers to the substrate layer was quite slow. In this way, the accumulated heat can provide a driving force that could promote the migration of β grain boundaries. As shown in Figure 6a, once the β grain boundaries were moved, the β grains in the higher layers tended to grow from the edge area of each layer. (3) The later-deposited layers had a higher temperature, which caused the preferential epitaxial growth of β grains. Furthermore, the preserved heat decreased the undercooling, which may promote the coarsening of β grains and the inside α’ phase. This outcome may explain why the size of the β grains in the upper layers was larger than that in the lower layers. The different α’ phase convexity shown in Figure 6b can also be ascribed to heat preservation. In addition, the heat gradients decreased as the distance from the center area to the outer area increased (Figure 11b).

As shown in Figure 11c, the distribution of the residual stress vector and the peak stress vector reached 1090 MPa. The maximum residual stress was identified near the first layer, and it was slightly lower than that of the single-bead part. This could be attributed to the upper layers’ re-heating, which decreased the residual stress to a certain extent. The calculated distortion of the substrate was 0.26 mm, which is just a bit higher than that of the single-bead part, indicating heat accumulation in the upper layers. The stress vectors were concentrated mainly in the first layer and the FZ with the increase in height, and the stress vector decreased and eventually could not be found. These results indicated that the lower layer generated greater residual stress that was caused by the high undercooling, which may explain the appearance of many smaller β grains in the lower layers (Figure 6a,e).

## 4. Conclusions

We conducted an experiment with double laser irradiation heads to investigate the microstructure of laser metal wire-deposited Ti6Al4V parts, and we performed a corresponding simulation with Simufact software. The regularities of the growth of the β grains and the newly precipitated α’ phase was described in addition to the reasons for the occurrence of the HAZ band in such α’ phase conditions. Our conclusions are as follows:

(1) The overlapping β grains within the single bead and lower layers in the thin-wall parts are ascribed to the special heat source and larger laser spot diameter during the printing process.

(2) The β grains displayed some regularities: (i) The lower the layer, the greater in number and the smaller the β grains were. (ii) The β grains showed a tendency to grow from the inner center area to the outer area in the single-bead parts and the lower layers of the thin-wall parts. (iii) β grains tended to grow in the edge area of each higher layer. (iv) Almost all the β grains in the upper layers were derived from the lower layers.

(3) A newly formed, precipitated α’ phase was observed.

(4) The white HAZ bands were ascribed to solute partitioning; however, since no β phase was detected, micro-segregation may be generated only between the α’ phase and α phase. The last two deposited layers did not show a HAZ band, while the microstructure of this region was more homogeneous. In addition, the position of the HAZ band in each layer could be changed. The microhardness was lower in the HAZ band region than on both the upper and lower sides.

However, the present study has its limitations: (i) the experiment mainly focuses on the discussion of the microstructure of as-built parts, and so the tensile test and other mechanical properties are not studied. The authors are going to deposit the block parts so that the dog bone tensile test samples could be fabricated. (ii) The transmission electron microscope (TEM) is considered to use for studying HAZ bands and microstructure further. (iii) In this simulation, the authors employed a single-laser heat source, which is different from the experiment wherein a double-laser heat source was used.

## Figures and Tables

**Figure 1 materials-15-07985-f001:**
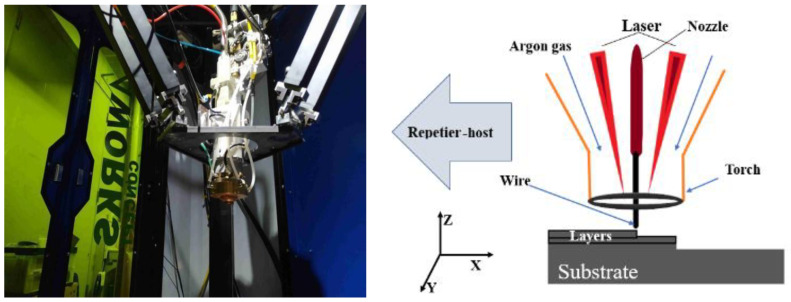
Coaxial printer setup and diagram.

**Figure 2 materials-15-07985-f002:**
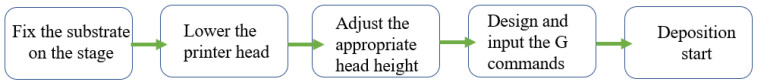
The preparation flow chart before printing.

**Figure 3 materials-15-07985-f003:**
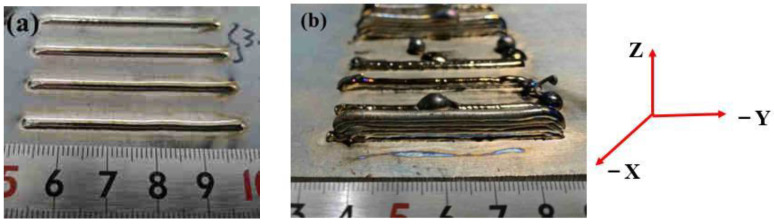
Single-bead (**a**) and single-pass seven-layer thin-wall (**b**) samples, along with the coordinates.

**Figure 4 materials-15-07985-f004:**
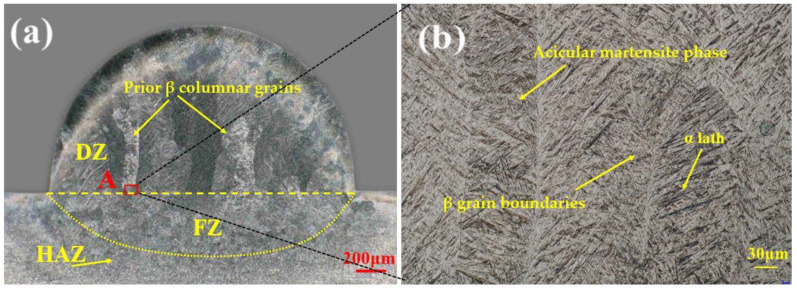
(**a**) The morphology of the cross-section of a typical etched single-bead Ti6Al4V sample, as shown by optical microscopy; (**b**) the area indicated in panel A.

**Figure 5 materials-15-07985-f005:**
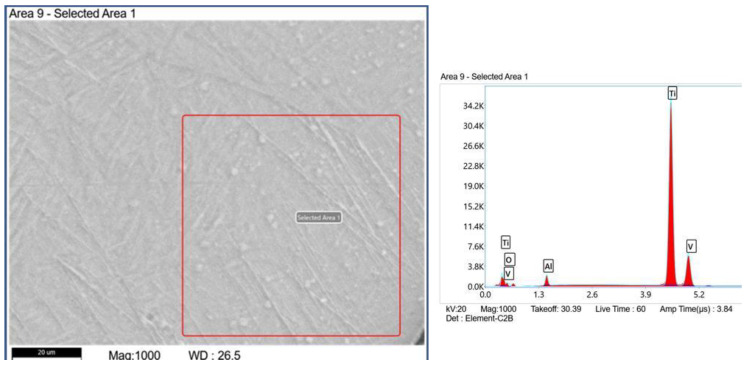
The EDS analysis in the edge area of the single-bead samples.

**Figure 6 materials-15-07985-f006:**
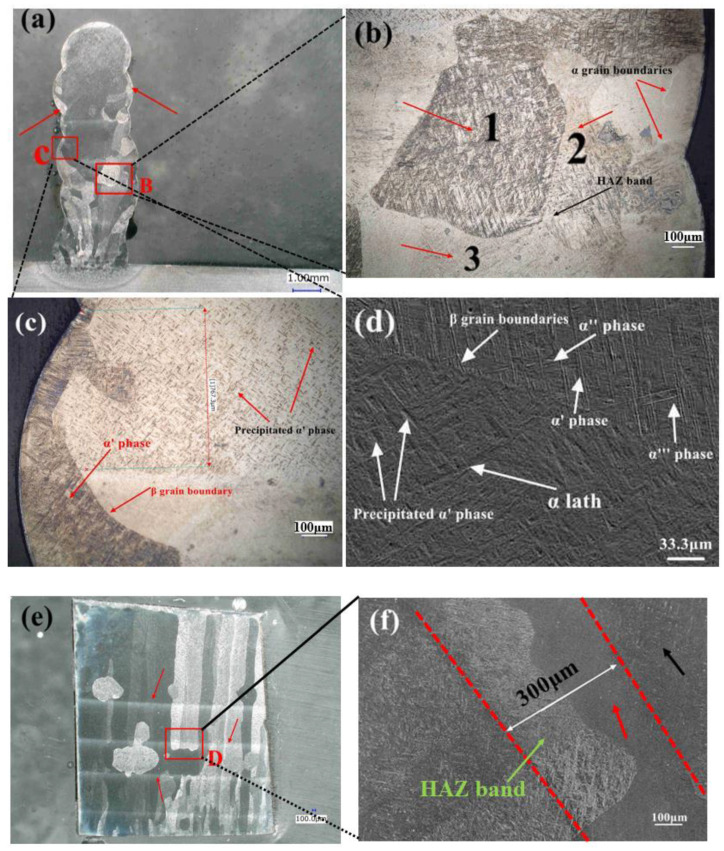
(**a**) An optical microscopy image of the cross-section cut from the X-Z plane. Red arrows indicate β grains at the edge area. (**b**) The area indicated in panel B of image (**a**). (**c**) The area indicated in panel C of image (**a**). (**d**) The microstructure of α’ and the precipitated α’ phase, as shown by the SEM. (**e**) The cross-section was cut from the second layer to the seventh layer of the Y-Z plane. (**f**) The area indicated in panel D in image (**e**) is revealed by the SEM. The red arrow indicates a transition area. The black arrow indicates the precipitated α’ phase.

**Figure 7 materials-15-07985-f007:**
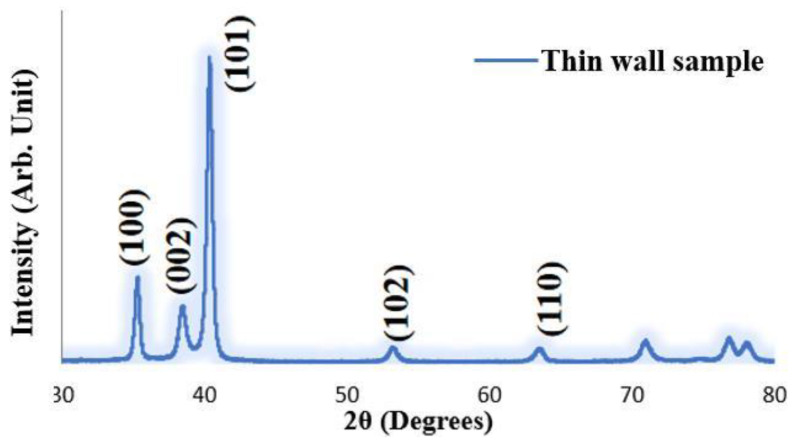
The X-ray diffraction pattern of the thin-wall as-built sample.

**Figure 8 materials-15-07985-f008:**
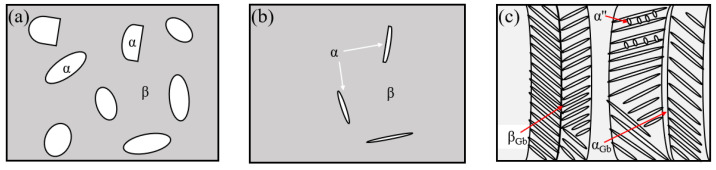
Schematic diagrams of martensitic transformation by LMwD. First layer: (**a**) room temperature, (**b**) above T_β_, and (**c**) cooling down below the M_S_. HAZ band formation: (**d**) the temperature rose to the T_β_, just in time. (**e**) Cooling down, below the M_S_.

**Figure 9 materials-15-07985-f009:**
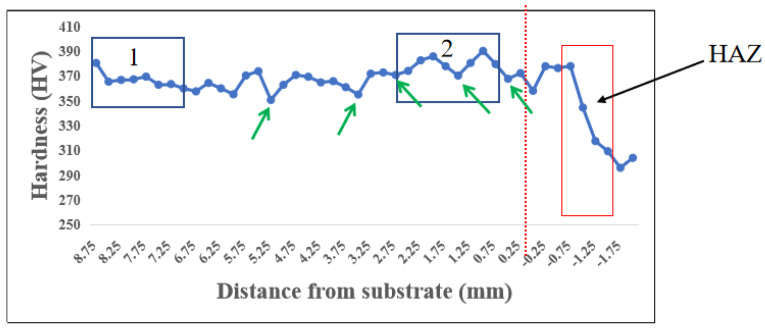
Microhardness of the cross-section of a thin-wall sample.

**Figure 10 materials-15-07985-f010:**
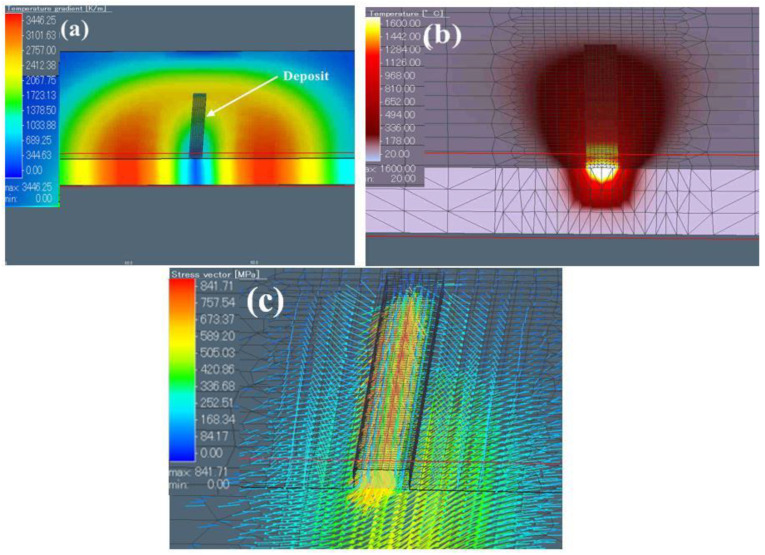
(**a**) The distribution of temperature, (**b**) the molten pool, and (**c**) the distribution of residual stress in a cross-section of the single-bead part.

**Figure 11 materials-15-07985-f011:**
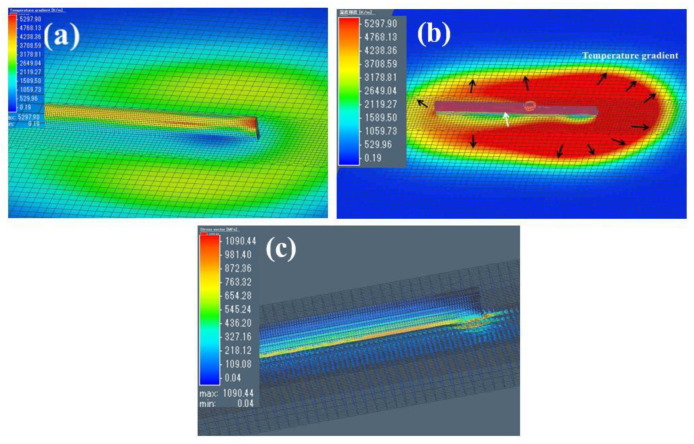
(**a**) The temperature distribution of the thin wall after printing. (**b**) The temperature gradients during the fifth layer’s deposition. (**c**) The residual stress vector and the maximum residual stress at room temperature.

**Table 1 materials-15-07985-t001:** Composition of the Ti6Al4V wire.

Element	O	N	C	Fe	Al	V	Ti
**wt %**	0.20	0.05	0.10	0.30	5.50~6.75	3.50~4.50	Bal.

**Table 2 materials-15-07985-t002:** Process parameters during printing.

Laser power P, W	1000
Laser spot dia., D*_w_*, mm	1.4
Head height, Z, mm	8
Head feed speed, F*_H_*, mm/min	500
Wire feed speed, F*_W_*, mm/min	191
Gas flow rate, L/min	20
Initial wire feed amount E, mm	5

**Table 3 materials-15-07985-t003:** The element composition of single-bead samples.

Position	Element	Weight, %	Atomic, %
Edge area of single-bead samples	Al	4.10	5.70
	Ti	78.70	60.70
	V	3.70	2.70
	O	13.40	31.00

**Table 4 materials-15-07985-t004:** The element composition near the HAZ band by EDS.

Position	Element	Weight, %	Atomic, %
Above HAZ band	Al	10.60	17.40
	Ti	85.10	78.60
	V	4.20	3.60
	O	-	-
HAZ band	Al	7.40	11.70
	Ti	85.90	76.50
	V	3.20	2.70
	O	3.40	9.20
Below HAZ band	Al	7.20	11.00
	Ti	82.80	71.30
	V	4.70	3.80
	O	5.40	13.80

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
