# Peer review of "Investigation of the Microstructure of Ti6Al4V Alloy by Coaxial Double Laser Metal-Wire Deposition"

_materials, 2022, doi:10.3390/ma15227985_

Round 1
Reviewer 1 Report
The authors studied the Investigation of the microstructure of Ti6Al4V alloy by coaxial double laser metal-wire deposition. The manuscript had an interesting topic and was well written, however it could only be accepted with the following minor revisions:
1. Shall avoid from using acronyms in keywords, e.g.: HAZ band.
2. Types of AM and its benefits/ limitations should be addressed.
3. The research gap of the study was not clearly described in the Introduction Section.
4. In order to prevent overlap features in Figure 2, please move or rescale the axis icon.
5. For section 2.2, it is recommended that the authors provide a flow chart for the experimental setup.
6. Results and discussion for sections 3.1, 3.2, and 3.4 were poor in citing/comparing to other or previous research findings (please improve).
7. It is suggested to add the limitation, future study, and develop implications for researchers in the conclusion section.
Reviewer 2 Report
The paper studies the microstructure of Ti6Al4V alloy by coaxial double laser metal-wire deposition. It is well written and structured. I have the following remarks:
1. There are differences in the notations of the same symbols in the table. 2 and in eq’s (1) and (2).
2. Improve the quality of fig.3. The use of yellow arrows is more appropriate here. This also applies to all other figures.
3. Please motivate the chosen numerical values of the parameters in table 2.
4. Can you compare the experimental and simulation results?
Reviewer 3 Report
The article is devoted to a topic that is very relevant and popular among researchers, and it is about the production of products from the Ti–6Al–4V alloy by additive manufacturing. It is especially interesting to get acquainted with studies where not a powder, but a titanium alloy wire was taken as a material.
The paper is written competently and clearly, well structured. During the research, modern scientific equipment and methods were used. However, when I got acquainted with the paper, I had a few comments and wishes to the authors.
1. The quality of photographs of microstructures shown in Figures 3 and 4 is very low. We need to increase their resolution.
2. In Figures 8 and 9, the temperature scale is not readable. You need to increase it and increase the resolution of the simulation results.
3. The work https://doi.org/10.3390/ma12193269 presents the results of similar studies, but with the use of powder rather than wire from the Ti–6Al–4V alloy. I think it is necessary to bring this article to References and compare the results.
4. Lines 214-216 state "The FZ, which is a transition area between the HAZ and the DZ, was filled with thinner equiaxial β grains compared to those in the DZ, indicating that higher residual stress existed". What is the basis for this assertion?
5. The paper says single-pass seven-layer thin-wall (b) sample soiled. In this case, it would be advisable to carry out Energy-dispersive X-ray spectroscopy (EDS) mapping of samples, as it is done in the article https://doi.org/10.3390/machines8040079. This would make it possible to show the distribution of chemical elements in different zones of a multilayer sample.
Round 2
Reviewer 1 Report
Author has made the changes according to reviewer comments. However, please do correction on the citation format in the line 536 (Ho et al. [25]).
Reviewer 2 Report
The paper is revised according to the reviews and is considerably improved. I reccomend publishing the paper as is.
Author Response
On behalf of all authors, I would like to express gratitude to the reviewer for your affirmation of our work.
Reviewer 3 Report
The authors of the article gave detailed responses to my comments and made corrections to the paper. I recommend publishing the article in this version.
Author Response

(The authors gave the same response as above.)
